# The Role of WNT Pathway Mutations in Cancer Development and an Overview of Therapeutic Options

**DOI:** 10.3390/cells12070990

**Published:** 2023-03-24

**Authors:** Wibke Groenewald, Anders H. Lund, David Michael Gay

**Affiliations:** Biotech Research and Innovation Centre, Faculty of Health and Medical Sciences, University of Copenhagen, 2200 Copenhagen, Denmark

**Keywords:** WNT, APC, β-catenin, ligand-dependent, colon cancer, endometrial cancer

## Abstract

It is well established that mutations in the canonical WNT-signalling pathway play a major role in various cancers. Critical to developing new therapeutic strategies is understanding which cancers are driven by WNT pathway activation and at what level these mutations occur within the pathway. Some cancers harbour mutations in genes whose protein products operate at the receptor level of the WNT pathway. For instance, tumours with *RNF43* or *RSPO* mutations, still require exogenous WNT ligands to drive WNT signalling (ligand-dependent mutations). Conversely, mutations within the cytoplasmic segment of the Wnt pathway, such as in *APC* and *CTNNB1*, lead to constitutive WNT pathway activation even in the absence of WNT ligands (ligand-independent). Here, we review the predominant driving mutations found in cancer that lead to WNT pathway activation, as well as explore some of the therapeutic interventions currently available against tumours harbouring either ligand-dependent or ligand-independent mutations. Finally, we discuss a potentially new therapeutic avenue by targeting the translational apparatus downstream from WNT signalling.

## 1. Introduction

WNT signalling is one of the most evolutionarily conserved signalling pathways and plays a critical role in many biological processes, including embryonic development, cell proliferation, self-renewal, and cellular differentiation. The first genes involved in the Wnt pathway were discovered in the 1980s. Firstly, the wingless (*wg*) gene was discovered as a critical regulator of segment polarity in *Drosophila* [1]. Soon after, the Int-1 oncogene was discovered to be activated in virally-induced breast tumours [2]. Int-1 was subsequently shown to be the mammalian homolog of *wg*, thereby being the first identification of the crossover of a developmental pathway with cancer [3].

*Wg*/Int-1 was shown to encode a secreted protein—known as WNT ligands, which are secreted in both an autocrine and paracrine fashion. To date, 19 WNT ligands have been identified [4]. These WNT ligands are approximately 40kDa lipid-modified glycoproteins that are palmitoylated by Porcupine (PORCN) [5] ahead of their transportation and secretion by the WNT Ligand Secretion Mediator (WLS) [6]. WNT ligands interact with one of ten Frizzled (FZD) receptors and an associated co-receptor, such as lipoprotein receptor-related proteins 5 and 6 (LRP5/6) [7]. Upon activation, FZDs recruit Dishevelled (DVL) to the cell membrane, which, in turn, can mediate downstream signalling [4]. It is important to note that WNT signalling is divided into canonical and noncanonical WNT signalling pathways [8]. Noncanonical WNT signalling results in the activation of the WNT-dependent calcium pathway or the planar cell polarity pathway [7]. This review focusses on the role of canonical WNT signalling and its therapeutic targeting in cancer.

Activation of the canonical WNT signalling pathway culminates in changes in gene expression driven by β-catenin-mediated transcription. β-catenin resides in three different locations: the cell membrane, where it is bound by E-cadherin; the cytoplasm, where it interacts with the destruction complex; and finally, the nucleus, where it drives transcriptional changes [9]. Of note, the vast majority of β-catenin resides in the membranous pool, bound to E-cadherin, forming a highly stable pool that is considered distinct from the ‘signalling’ pool of β-catenin [10]. The gene encoding β-catenin, *CTNNB1*, is constantly transcribed and in the absence of a WNT-ligand signal, the protein product is marked by the destruction complex for subsequent proteasomal degradation in the cytoplasm [10]. The destruction complex consists of two scaffold proteins, AXIN and APC, as well as two kinases, GSK3β and CK1α [11]. APC is a scaffold protein that binds β-catenin and AXIN. AXIN interacts with GSK3β and CK1α, as well as Dishevelled. CK1α phosphorylates β-catenin at Ser45 (serine 45), allowing for the subsequent phosphorylation of Ser33, Ser37, and Thr41 (threonine 41) by GSK3β [10]. Upon its phosphorylation, β-catenin is recognised by the ubiquitin ligase β-TrCP, which ubiquitinates phosphorylated β-catenin marking it for proteasomal degradation [12]. However, in the presence of WNT signalling, the destruction complex is recruited to the cell membrane via the interaction of AXIN and Dishevelled following the binding of a WNT ligand to FZD-LRP5/6 co-receptor complexes [13]. Whilst β-catenin can still be phosphorylated, it can no longer be ubiquitinated by β-TrCP as it is dissociated from the destruction complex [10]. Consequently, the destruction complex becomes saturated with phospho-β-catenin, therefore *de novo* synthesized β-catenin can translocate to the nucleus to mediate transcriptional changes [10].

Once in the nucleus, β-catenin binds to two transcription factors, Lymphoid Enhancer Factor-1 (LEF1) and T-cell factor (TCF), displacing the transcriptional repressor Groucho, thereby allowing for binding to WNT-responsive DNA elements [11]. Additional transcriptional co-activators of β-catenin, such as B-cell lymphoma 9 (BCL9), B-cell lymphoma 9-like (BCL9l), and Pygopus (PYGO), cooperate in β-catenin-mediated transcription, interacting with the N-terminal domain of β-catenin, forming part of the WNT enhanceosome [14]. Several chromatin-modifying enzymes also interact with β-catenin via its C-terminal domain, such as CREB binding protein (CBP) and BRG1 [15]. Several WNT target genes are known to be associated with cell growth and proliferation. Some of the best known WNT target genes include *MYC* [16], which encodes a transcription factor that controls the expression of many proliferative genes, and *CCND1* [17], which encodes Cyclin D1 [17], an important regulator of the cell cycle. Hence, dysregulated WNT signalling can drive proliferative transcriptional programmes, thereby leading to aberrant proliferation.

Several mechanisms serve to further modify the WNT response through either negative feedback loops or potentiation of Wnt signalling. Among these is the family of roof-plate specific spondins (R-spondins) 1–4, which are secreted ligands for the leucine-rich repeat-containing G-protein coupled receptors (LGR) 4–6 [18]. R-spondin ligands form a complex between LGRs and the transmembrane E3 ubiquitin ligases zinc and ring finger 3 (ZNRF3) or ring finger protein 43 (RNF43), mediating the internalization of this receptor complex. In the absence of an R-spondin ligand, ZNRF3/RNF43 ubiquitinate Frizzled receptors, marking them for internalisation and degradation [18]. Hence, these ubiquitin ligases, which are also WNT target genes, serve as part of a negative feedback loop to limit WNT signalling. However, *LGR5* is also a WNT target gene; therefore, the secretion of R-spondins can potentiate Wnt signalling by inhibiting ZNRF3/RNF43 [19]. Indeed, it is worth noting that whilst all four R-spondins can bind to LGRs, there is over a 100-fold difference in their EC_50_ for receptor activation, with RSPO2 and RSPO3 being the most potent [20].

Similar to ZNRF3/RNF43, there are other WNT target genes whose protein products form negative feedback loops. Firstly, the destruction complex scaffold proteins AXIN1/2 are WNT target genes; hence, the incorporation of newly synthesized AXIN1/2 into the destruction complex destabilizes β-catenin and dampens the WNT signal [21]. Another WNT target gene, *DKK1*, encodes the secreted protein Dickkopf-related protein 1. DKK1 inhibits the dimerization of LRP5/6 with Frizzled receptors, thereby inhibiting Wnt signalling [22]. Another secreted factor, NOTUM, is a palmitoleoyl-protein carboxylesterase that removes the essential palmitoleate moiety from WNT ligands, thereby inhibiting their activity [23]. NOTUM has been shown to be upregulated in *Apc*-deficient mouse intestinal tumours, indicating that it could be a WNT target gene [24]. An overview of the canonical WNT pathway and its feedback mechanisms is summarized in Figure 1.

## 2. Wnt Activating Mutations in Cancer

It is well established that mutations in components of the canonical WNT signalling pathway play a major role in several cancers [4,11]. However, it is important to note that there is a degree of specificity as to which WNT pathway genes are mutated in certain cancers, as summarized in Figure 2. For instance, truncating mutations in *APC* are found in approximately 70% of colorectal cancer (CRC) patients, whilst activating mutations in *CTNNB1* are far less frequent in this disease [9]. Conversely, *CTNNB1* is mutated in approximately 25% of patients with hepatocellular carcinoma (HCC), yet *APC* is rarely mutated in these patients, see Figure 2. While the precise reasons behind this tissue specificity remain unclear, it has been proposed to be dependent on the homeostatic levels of WNT signalling, which can play a role in determining the fitness of mutant cells relative to their normal neighbours [4]. One explanation as to why activating *CTNNB1* mutations are rare in CRC is due to the high levels of E-cadherin expression in the colonic epithelium that serves to sequester β-catenin to the cell membrane and therefore limits the transforming properties of *CTNNB1* mutations in CRC compared to *APC* mutations [9].

Both *APC* and *CTNNB1* are mutated in approximately 12% and 30% of endometrial cancers, respectively [25]. However, only half of these *APC* mutations give rise to truncated proteins; therefore, *CTNNB1* mutations are likely to be the dominant WNT-activating mutation in endometrial cancer [26]. Interestingly, loss of *Apc* alone did not drive malignant transformation in an endometrial cancer mouse model unless combined with concomitant *Pten* loss [27], thereby suggesting that not all WNT pathway mutations are equal in their transforming potential in endometrial cancer. It is important to note that in this review we solely focus on DNA mutations in WNT pathway genes, leading to altered protein function. However, there is evidence of epigenetic deregulation of the WNT pathway in several cancers too. For instance, the WNT antagonist DKK1 has been shown to be epigenetically downregulated in CRC [28]. Moreover, genes encoding another type of WNT antagonist, Secreted Frizzled-Related Proteins, have also been shown to be epigenetically inactivated in gastric cancer [29].

**Figure 2 cells-12-00990-f002:**
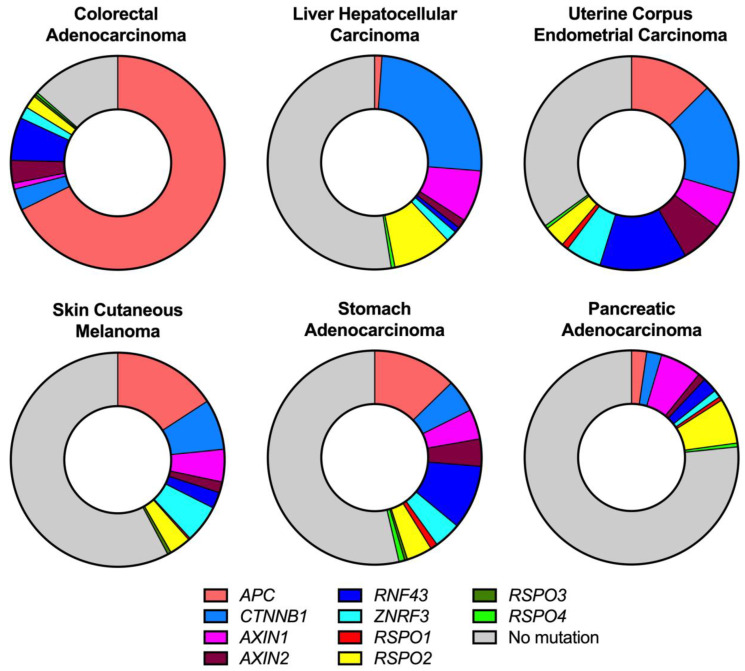
An overview of the frequency of certain WNT pathway mutations found in colon, liver, endometrial, gastric, pancreatic, and skin cancer. Data from The Cancer Genome Atlas’s PanCancer Atlas on cBioportal [30,31], as of 15 February 2023. No mutation indicates patients in the queried datasets that did not have a mutation in any of the genes shown above.

### 2.1. APC Mutations

The APC protein is a large scaffold protein containing multiple domains required to interact with its binding partners in the destruction complex and β-catenin. For instance, the three 15 amino-acid repeats (15AARs) and 20 amino-acid repeats (20AARs) bind β-catenin, whilst three Ser-Ala-Met-Pro (SAMP) repeats bind AXIN [32]. In colon cancer, the vast majority of mutations are found before the 5′ end of exon 15 and give rise to truncated proteins. There is a clearly defined hotspot for mutations known as the Mutation Cluster Region (MCR), see Figure 3A [33]. The resulting truncated proteins typically only retain 1–3 intact 20AARs, whilst the SAMP motifs are lost, leading to aberrant, but not maximal, WNT signalling [33]. Hence, the dominant *APC* mutation found in CRC still retains some β-catenin binding capacity and therefore does not lead to maximal WNT signalling. This has led to the ‘just right WNT signalling hypothesis’, which suggests that a certain level of WNT signalling in CRC is optimal for transformation [34]. For instance, there is evidence that the number of retained 20AARs reflects CRC tumour location: proximal colonic tumours retain more 20AARs than distal colonic tumours and therefore likely have differing pathological levels of WNT signalling [33]. This tumour distribution could be influenced by the decreasing WNT gradient that runs from the proximal to distal colon [35]. It is proposed that high basal WNT signalling in the proximal colon is not favourable for tumours that have high pathological WNT signalling; consequently, distal colonic tumourigenesis is favoured for these tumours [35]. There is evidence from mouse models of intestinal tumourigenesis that support this hypothesis. A decreasing WNT gradient runs from the proximal to distal end of the mouse small intestine [35]. The *Apc*^Min/+^ mouse harbours germline mutations in *Apc* that encode a truncated APC protein lacking any β-catenin binding capacity and typically gives rise to tumours in the distal small intestine [36,37]. The *Apc*^1322T/+^ mouse also has a germline *Apc* mutation but retains one 20AAR and typically develops proximal intestinal tumours with lower levels of nuclear β-catenin compared to *Apc*^Min/+^ mice [37]. Hence, the lower levels of Wnt signalling in *Apc*^1322T/+^ tumours are compatible with tumour formation in the higher basal WNT levels of the proximal small intestine. Beyond colon cancer, *APC* is also mutated in around 13–15% of uterine endometrial, stomach cancer, and skin cutaneous melanoma (Figure 2).

### 2.2. CTNNB1 Mutations

Whilst *APC* mutations give rise to truncated proteins, particularly in CRC, *CTNNB1*-activating mutations give rise to stabilized β-catenin through point mutations in HCC, endometrial cancer, and pancreatic cancer. There is a hotspot of point mutations in exon 3 of *CTNNB1*, clustered around the different phosphorylation sites targeted by the destruction complex [26]. The actual phosphorylation sites Ser33, Ser37, Thr41, and Ser45 are mutated such that they can no longer be phosphorylated by CK1α or GSK3β, thus stabilizing β-catenin. Additionally, Asp32 and Gly34 are frequently mutated, and these sites are required for correct β-TrCP binding; therefore, following their mutation, β-catenin ubiquitination is impaired [26]. The mutational hotspot in exon 3 of *CTNNB1* is summarized in Figure 3B. Outside this mutation cluster in exon 3, two less frequent point mutations, N387K and K335I/T, have been identified in HCC patients [40]. These two point mutations are found within the armadillo repeats of β-catenin, which are involved in the binding of β-catenin to its various binding partners, including APC and AXIN. Interestingly, whilst the aforementioned mutations can lead to β-catenin stabilization, they activate the WNT pathway to different degrees. For instance, the two mutations outside of the exon 3 cluster, N387K and K335I/T, showed the weakest signal in a WNT reporter assay compared to Ser33, Thr41, and Ser45 mutations [40]. Similarly, whilst Ser45 gave a weaker signal in a WNT reporter assay compared to Thr41 and Ser33 mutations, this *CTNNB1* mutation was frequently duplicated in HCC, leading to high β-catenin activity [40]. The same recurrent *CTNNB1* exon 3 mutations are also found in endometrial cancer; however, there was no correlation between the levels of nuclear β-catenin and specific *CTNNB1* mutations [41].

### 2.3. AXIN Mutations

Both AXIN1 and AXIN2 can serve as scaffold proteins in the destruction complex to regulate β-catenin levels. *AXIN1* is mutated in approximately 8% of HCC patients [42] and deletion of *Axin1* in the murine liver can give rise to hepatic tumours [43]. Both *AXIN1* and *AXIN2* are mutated in approximately 14% of uterine endometrial cancers and *AXIN2* is mutated in about 5% of CRC patients (Figure 2). Whilst there is a high degree of homology between these two proteins, their mutation patterns found in cancer are markedly different. For instance, *AXIN1* is mutated throughout the entire coding sequence, with different mutations between tumour types. *AXIN2*, on the other hand, has a recurring frameshift mutation in exon 7 across multiple tumour types [44], particularly in CRC and stomach cancer, see Figure 3C. This gives rise to a truncated protein, stabilized β-catenin, and activated WNT signalling.

### 2.4. RNF43/ZNRF3 Mutations

The two ubiquitin ligases RNF43 and ZNRF3 are part of a negative feedback loop that results in the internalization and degradation of Frizzled receptors, thereby reducing WNT signalling. *RNF43* is mutated in uterine, colorectal, pancreatic, stomach, and skin cancer. In both CRC and endometrial cancer, a hotspot mutation at G659 induces a frameshift, leading to a truncated form of RNF43 [45]. The same study also identified a mutually exclusive relationship between mutations in *RNF43* and *APC* in CRC patients, suggesting an activating role of truncating mutations in *RNF43* in the WNT pathway of colon tumours [45]. The same G659 frameshift mutation accounts for approximately 25% of *RNF43* mutations in gastric cancer [46]. Of note, this frameshift mutation is not predicted to lead to altered protein functionality [47]. Furthermore, expression of G659fs RNF43 did not affect β-catenin signalling in colorectal cancer cells [48]. The authors also showed that N-terminal truncating mutations in *RNF43* are required to drive increased β-catenin signalling; hence, further work is needed to understand the role of G659fs *RNF43* mutations in certain WNT-driven cancers [48]. Interestingly, *RNF43* mutations in gastric cancer, endometrial cancer, and CRC are associated with microsatellite instable tumours [45,46]. Mutations in *ZNRF3* are found in both uterine and skin cancers, but have been studied less frequently than *RNF43* mutations.

### 2.5. RSPO Mutations

There are four members of the RSPONDIN family, of which *RSPO2* is most frequently mutated in cancer. *RSPO2* and *RSPO3* mutations are found in approximately 10% of colon cancers and occur in a mutually exclusive manner with *APC* mutations [49]. The study identified recurring *PTPRK-RSPO3* and *EIF3E-RSPO2* fusion mutations [49], and *PIEZO1-RSPO2* fusion mutations have also been identified in CRC [50]. As a result of these fusion mutations, high levels of *RSPO2* and *RSPO*3 are detected in tumours harbouring these mutations [50,51], and indeed, mice harbouring *Rspo2* or *Rspo3* fusion mutations develop intestinal tumours [52]. Recurrent *RSPO2* mutations have also been identified in HCC as a result of a large deletion of chromosome 8q23.1, and whilst the authors speculate a similar *EIF3E-RPSO2* mutation to that found in CRC, they were unable to verify an *EIF3E-RSPO2* amplicon due to DNA and RNA fragmentation in the formalin-fixed paraffin-embedded samples used [53]. High levels of *RSPO2* are detected in these tumours, whilst immunohistochemical analysis of HCC tumours harbouring *RSPO2* mutations showed both strong nuclear β-catenin and glutamine synthetase staining, indicating a strong WNT pathway activation in these tumours [53]. *RSPO2* fusion mutations have also been reported in both gastric [54] and prostate cancer [55]; however, all four RSPONDIN proteins have been shown to be overexpressed across many other types [56].

## 3. Therapeutic Options against Aberrant WNT Signalling in Cancer

Even though the WNT pathway activating mutations outlined above ultimately lead to increased levels of nuclear β-catenin, the actual dependency of these mutations on the WNT ligands themselves to drive increased WNT signalling is determined by their position within the signalling cascade [21]. As such, certain mutations can be defined as Wnt ligand-independent, such as mutations in *APC*, *CTNNB1,* and other components of the destruction complex. However, other driver mutations, such as in *RNF43* or *RSPO2*, are typically WNT ligand-dependent mutations. These two classifications have important implications for potential treatment options for patients when using WNT-targeting-based therapies. We will now discuss some of the therapeutic options for use against certain WNT-activating mutations and at different levels of the WNT pathway [57]. An overview of WNT-targeting therapies is shown in Figure 4.

### 3.1. WNT Ligand/Receptor-Based Therapies

#### 3.1.1. Porcupine Inhibitors

Currently, there are no WNT-targeting therapies approved for cancer treatment; however, a number of potential therapies targeting WNT ligands or the receptor complex have entered clinical trials for a number of cancers, further reviewed in [58]. There has been a strong focus on inhibiting WNT ligand secretion through PORCN inhibition, with a number of PORCN inhibitors (PORCNi) in clinical trials. WNT ligands are palmitoylated at Ser209 by PORCN ahead of their secretion [5]. The PORCN inhibitor WNT974 (LGK974) has been shown to inhibit the palmitoylation of all tested canonical WNT ligands in vitro and was well-tolerated in WNT-dependent tissues in vivo [59]. Subsequent studies have shown WNT974 to be particularly effective in cancer models harbouring either *RNF43* or *RSPO* mutations. For instance, WNT974 treatment inhibited the growth of *RNF43*-mutant pancreatic tumours in a xenograft model [60]. Additionally, WNT974 treatment regressed murine intestinal tumours harbouring either *Rspo2* or *Rspo3* fusion mutations [52], whilst another PORCNi suppressed the growth of *RSPO3*-mutant patient-derived colon cancer organoids in a xenograft model [61]. Another study showed that treatment with a different PORCNi, Wnt-C59, prevented neoplastic growth of *Rnf43* and *Znf3* mutant intestinal cells [62]. Additionally, in a WNT-driven model of breast cancer, Wnt-C59 suppressed the progression of mammary tumours in MMTV-WNT1 transgenic mice and downregulated WNT target genes [63]. However, PORCNi treatment did not reduce proliferation in intestinal crypts harbouring homozygous *Apc* mutations [64], suggesting that cancers that harbour mutations in the destruction complex or activating mutations in *CTNNB1* are likely to be insensitive to *PORCNi* treatment since these mutations activate the Wnt pathway downstream of the Wnt receptor complexes. It is important to note that whilst there have been promising results treating *RNF43* or *RSPO2/3* mutant cancers with porcupine inhibitors, one on-target side effect of these drugs is a loss of bone density since WNT signalling plays an important role in bone homeostasis [65]. However, attempts are underway to mitigate bone loss in response to PORCNi treatment through concurrent treatment with drugs used to treat osteoporosis [66].

#### 3.1.2. WNT Ligand and Receptor Targeting Strategies

An analogous strategy for blocking WNT ligand secretion involves blocking the binding of WNT ligands to their FZD receptors. For instance, OMP-54F28 (Ipafricet) is a truncated FZD8 receptor fused to an antibody that can bind free WNT ligands, thereby blocking WNT signalling. It has been shown to block WNT signalling and yielded promising results in several pre-clinical cancer models [67]. Alternative strategies have sought to block FZD receptors on the cell surface, thereby preventing binding of their cognate WNT ligands. One such example is OMP-18R5 (Vantictumab), which binds to Frizzled receptors and inhibits downstream WNT signalling [68]. OMP-18R5 showed promising efficacy as a single agent or in combination with other chemotherapies in xenograft models of breast, pancreatic, lung, and colon cancer [68]. Whilst information is lacking on the mutational status of all xenograft models, the authors stated that the colon cancer model was wildtype for both *APC* and *CTNNB1* [68]. It is also important to note that OMP-18R5 did not inhibit WNT signalling following intracellular pathway activation by concomitant treatment with a GSK3β inhibitor [68]. This suggests that cancer cells harbouring mutations in components of the destruction complex or *CTNNB1* would likely be insensitive to OMP-18R5 treatment. However, this is conflicting evidence for this view, since OMP-18R5 treatment reduced both growth and WNT target gene expression in *Apc*-null gastric cancer organoids [69]. Hence, further studies are required to assess the efficacy of OMP-18R5 in specific mutational spectra.

#### 3.1.3. Other Extracellular Targeting Strategies

Finally, several attempts have been made to target R-Spondin signalling in certain WNT-driven cancers. The anti-RSPO3 antibody OMP-131R10 (Rosmantuzumab) is currently being tested in several clinical trials. However, we can gain some indication of the efficacy of this treatment in specific mutational backgrounds based on a previous study. Storm and colleagues showed that anti-RPSO3 antibodies inhibited the tumour growth of *PTPRK-RSPO3* fusion mutant colorectal cancer patient-derived xenografts (PDXs) [70]. However, the authors also showed that *APC*-mutant PDXs are resistant to this treatment [70], thereby highlighting the necessity to tailor WNT-targeting therapies to certain mutational profiles. Finally, another potential therapy option is likely relevant to colorectal cancer patients diagnosed with Familial Adenomatous Polyposis (FAP) who harbour germline *APC* mutations. *Apc*-null intestinal cells have been shown to secrete NOTUM, a WNT deacylase, in a manner that suppresses WNT signalling in neighbouring WT cells [24]. This significantly increased the clonal capacity of these *Apc*-mutant cells and their ability to repopulate the crypts of the mouse intestine, which is a precursor to tumour formation. Indeed, deletion of *Notum* throughout the entire murine intestinal and colonic epithelia significantly extended the survival of *Apc*^Min/+^ mice, which harbour germline *Apc* mutations [24]. Interestingly *Rnf43* and *Znrf3* null intestinal cells did not express NOTUM [24], further highlighting that as well as having different sensitivities to therapies, ligand-dependent and ligand-independent tumours have differing transcriptional responses following WNT-pathway activation. Finally, the authors showed that NOTUM inhibitor treatment reduced the clonal capacity of *Apc*-null intestinal cells and reduced the number of tumour lesions in an *Apc*-mutant-driven model of intestinal tumourigenesis at a 21-day timepoint [24]. Therefore, NOTUM inhibitor treatment could provide an efficacious strategy for limiting tumour outgrowth in FAP patients who are predisposed to CRC.

### 3.2. β-Catenin Targeting Therapies

The previous section focussed on therapies targeting the extracellular components of WNT signalling. These therapies have largely proven most effective in ligand-dependent pre-clinical models but have shown limited efficacy in ligand-independent cancer models, typically due to constitutively active WNT signalling arising from mutations in WNT pathway components that operate downstream of Frizzled receptors, such as in *CTNNB1* and *APC*. Now, we review some of the therapies that target β-catenin itself, as well as a potential therapeutic option downstream of WNT signalling.

#### 3.2.1. Targeting β-Catenin in the Destruction Complex

A number of strategies have sought to boost the activity of the destruction complex as a means to reduce levels of β-catenin. For instance, AXIN is poly-ADP-ribosylated by Tankyrase, marking it for degradation via the ubiquitin–proteasome pathway [71]. Consequently, Tankyrase inhibitors have been developed to stabilize AXIN levels and therefore antagonize WNT signalling [71]. Tankyrase inhibitors have shown promising efficacy in both colon cancer cell lines and colon cancer mouse models that harbour *Apc* mutations [72,73]. However, one potential caveat of Tankyrase inhibitors is their toxicity within the intestine [73] and therefore their use in a clinical setting is likely limited.

Similar to Tankyrase inhibitors, Pyrvinium has been used to increase the activity of the destruction complex by binding to and inducing a conformational change in CK1α, promoting its kinase activity [74]. Pyrvinium treatment has shown efficacy against colon cancer cell lines, both in vitro and in xenograft models [75,76]. Moreover, Pyrvinium treatment reduced the number of intestinal adenomas and the expression of WNT target genes in *Apc*^Min/+^ mice after 10 weeks of treatment [77]. Finally, Pyrvinium treatment impaired the growth of HCC cell lines in vitro as well as tumour growth in an HCC xenograft model [78]. Importantly, a phase I clinical trial is currently recruiting patients to test the safety and tolerance of Pyrvinium (NCT05055323) in pancreatic cancer. Altogether, these studies highlight the potential of targeting WNT signalling at the level of the destruction complex.

#### 3.2.2. Targeting β-Catenin-Mediated Transcription

Downstream of the destruction complex, there have been several attempts to target β-catenin by disrupting its interactions with transcriptional co-activators. For instance, to generate an active transcriptional complex, β-catenin interacts with CBP and p300, which are histone acetyltransferases [15]. The small molecule ICG-001 has been shown to disrupt the interaction of β-catenin with CBP. ICG-001 treatment lowered WNT target gene expression in colon cancer cell lines harbouring *APC* mutations and reduced tumour growth in a colon cancer xenograft model [79]. Importantly, given the previously mentioned on-target toxicity of some WNT-targeting therapies, including PORCNi and Tankyrase inhibitors, the authors showed that ICG-001 only induced apoptosis in cancer cells and not in normal cells in their xenograft model [79]. A closely related compound of ICG-001, PRI-724, reduced proliferation and WNT target gene expression, whilst inducing apoptosis in HCC cell lines [80]. PRI-724 has successfully undergone phase I clinical trials in combination with Gemcitabine in advanced pancreatic adenocarcinoma patients and was deemed safe for further clinical testing (NCT01764477).

Several genetic studies have shown that targeting components of the WNT enhanceosome, specifically, BCL9, BLC9l, and Pygo, could represent a promising therapeutic target. For instance, deletion *of Bcl9* and *Bcl9l* or *Pygo1* and *Pygo2* significantly extended survival in both *Apc*^Min/+^ and *Apc*^1322T/+^ mouse models of colon cancer [81]. Furthermore, concomitant deletion *of Bcl9* and *Bcl9l* with *Apc* significantly reduced proliferation and WNT target gene expression in murine intestinal crypts compared to crypts harbouring homozygous deletion of APC [82]. Likewise, deletion of both *Bcl9* and *Blc9l* significantly extended the survival of mice that harboured hepatic *Ctnnb1* mutations and reduced WNT target gene expression in hepatocytes [82]. Moreover, deletion of *Pygo2* significantly delayed tumour onset in a WNT-driven mammary tumour mouse model [83]. Together, these studies suggest that both BCL9/9l and Pygo1/2 are attractive therapeutic targets in cancers that harbour mutations in both *APC* and *CTNNB1*. Indeed, a BCL9 peptide mimetic was shown to disrupt native β-catenin/BCL9 complexes and lead to reduced WNT target gene expression in colon cancer cell lines and inhibited tumour growth in a colon cancer xenograft model [84] Other small molecule inhibitors of BCL9 have also been discovered. For instance, carnosic acid, a naturally occurring compound found in rosemary, reduces WNT signalling in cell lines and the mouse epithelium, but also reduces tumour burden in *Apc*^Min/+^ mice [85]. More recently, E722-2648 was identified in a high-throughput screen for inhibitors of BCL9’s interaction with β-catenin and was shown to reduce WNT target gene expression in human colon cancer organoids and reduce tumour growth in a colon cancer xenograft model [86] Hence, there are promising BCL9 inhibitors that have shown efficacy in various pre-clinical settings and could pave the way for novel treatments in WNT-driven cancers. To date, no inhibitors of Pygopus have been identified.

## 4. A Therapeutic Option beyond WNT Signalling

Thus far, we have discussed therapeutic approaches that target the inhibition of ligand-dependent and -independent WNT activation to inhibit β-catenin-mediated expression of WNT target genes. However, additional opportunities to treat WNT-driven cancers may lie downstream of the pathway itself. In this section, we explore new possibilities for intervening with the effects of WNT pathway activation, focusing mainly on the downstream effects on the dysregulation of protein synthesis. Deregulated translation is a hallmark of many cancers [87]; however, it could be especially relevant in certain WNT-driven cancers, particularly colorectal cancer [88].

### 4.1. The Key Players in Translation and Their Links to WNT Signalling

The translation of mRNA into protein is mediated by ribosomes and numerous auxiliary proteins, termed translation factors. Of the latter, briefly, initiation factors play a role in assembling translation elongation competent ribosomes at the start codon of the mRNA to be translated [89]. Elongation factors, on the other hand, play a critical role after translation has begun through the recruitment of tRNAs charged with amino acids to the ribosome, as it proceeds along the mRNA, synthesizing a growing peptide chain [90]. The roles of translation initiation and elongation factors in cancer are further reviewed in [91,92,93]. The mammalian ribosome is a staggeringly complex molecular machine, which, in its core, consists of four ribosomal RNAs (rRNAs) and approximately 80 ribosomal proteins (RPs) arranged into a large and a small subunit [94].

One of the major transcriptional targets of the canonical WNT signalling pathway is the protooncogene *c-Myc* [16,95]. The *c-Myc* gene encodes the transcription factor MYC, which is a master regulator of ribosome biogenesis and transcription of certain translation initiation factors, such as eIF2α, eIF4E, and eIF5A [96,97,98]. Indeed, MYC is known to bind to upstream elements of ribosomal DNA genes (encoding rRNA) and induce the activity of RNA polymerase I for the transcription of rRNA precursors, thereby driving ribosome biogenesis [99,100]. Moreover, MYC is known to drive the transcription of many ribosomal protein genes that are typically deregulated in various cancers [101]. It is interesting to note that the patient subtype in CRC typified by *APC* mutations, along with high WNT and MYC target gene expression, also shows an enrichment of the ‘translation ribosome’ signatures compared to other CRC subtypes [102]. Indeed, CRC has been proposed to be a cancer that is ‘addicted to translation’ [88]. Mouse models of CRC have shown that global translation rates are significantly upregulated in *Apc*-deficient tumours and cell lines [103,104]. Several studies have shown promising efficacy of targeting both translation initiation and elongation via genetic and chemical approaches in mouse models of colon cancer [104,105,106,107]. Aside from targeting translation factors, haploinsufficiency of *Rpl24*, a ribosomal protein gene, suppressed tumour growth in an *Apc* and *Kras* mutant mouse model of colorectal cancer [108]. Interestingly, the same study showed significant translational upregulation of many ribosomal protein genes following *Apc* deletion in the murine intestine, further supporting the idea that WNT activation drives upregulation of the translation machinery [108]. Indeed, the overexpression of many ribosomal proteins has been reported in CRC, as well as in several other cancer types [109]. Finally, inhibition of WNT signalling in both colon cancer cell lines and mouse organoids leads to a reduction in rRNA biosynthesis and global protein synthesis rates [110]. The authors also showed, in CRC PDX models, that cells with the highest EPHB2 expression, a known WNT target gene, displayed the highest rates of both protein synthesis and rRNA synthesis, indicating high rates of ribosome biogenesis and translation in WNT-high tumour cells [110].

Focussing on other cancer types, a study identified a negative correlation between *APC* expression and the expression levels of components of the translational apparatus in both breast and lung cancer [88]. Low levels of *APC* expression correlated with high expression of the translation machinery but also increased sensitivity to oxaliplatin through targeting of rRNA synthesis and ribosome biogenesis [88]. Treatment of an *RNF43*-mutant pancreatic cancer cell line with a PORCNi led to a pronounced decrease in the expression of ribosomal protein genes and ribosome biogenesis genes [111]. Finally, inhibition of β-catenin in triple negative breast cancer cell lines was shown to impair ribosome biogenesis, and the authors moreover showed that a number of genes encoding ribosome biogenesis factors were predicted to be TCF/LEF targets, including LAS1-like ribosome biogenesis factor (LAS1L), an endonuclease involved in pre-rRNA processing, and Fibrillarin, a methyltransferase that mediates 2′-O-methylation of rRNA [112]. Altogether, these studies indicate that both translation and ribosome biogenesis are downstream targets of the WNT pathway in certain cancers, as summarized in Figure 5.

### 4.2. Are Ribosomes Altered in Cancer?

Ribosomes have long been viewed as a homogenous population that mediates the translation of mRNA into protein, with no inherent ability to regulate translation. However, this view is changing with growing evidence of ribosome heterogeneity, which can emanate from several sources, as reviewed in depth in [94,113,114]. At the ribosomal protein level, several studies have shown that the stoichiometry of certain ribosomal proteins is altered in different cell lines [115] or that different ribosomal paralogs can be incorporated into ribosomes [116]. For example, ribosomes containing ribosomal protein RPL10a have been shown to favour the translation of WNT-pathway mRNAs in the developing mouse embryo [117]. Whilst there is currently limited knowledge regarding differences in ribosome protein stoichiometry or ribosome protein paralog switching in solid cancers, a number of studies have identified mutant ribosomal proteins in leukaemia [118,119]. Moreover, dysregulated expression of certain ribosomal proteins has been associated with tumour progression and therapeutic resistance across a variety of cancers [120].

At the rRNA level, the mammalian genome contains many different rDNA alleles, which are known to give rise to variant rRNA sequences in different tissue types, but also between individuals [121]. More recently, a study has shown changes in rRNA sequences between healthy and cancerous biopsy samples [122] Ribosomal RNA is extensively post-transcriptionally modified, with 2′-O-methylation and pseudouridylation being the most abundant types of modification that can vary between cancer cell lines and tissues, giving rise to heterogeneous ribosomes [94]. For example, differential rRNA 2′-O-methylation patterns have been identified in a cohort of human breast cancer patients and in patients with diffuse large B cell lymphoma [123,124]. Moreover, our lab has shown that manipulation of a single MYC-induced rRNA 2′-O-methylation alters the translational output of HeLa cells and reduces their proliferation rate [125]. It is interesting to note that deletion of C/D box small nucleolar RNAs (SNORDs), which guide specific rRNA 2′-O-methylation by fibrillarin, affected the sensitivity of *S. cerevisiae* to ribosome-specific antibiotics [126]. This suggests that changes in 2′-O-methylation may influence ribosome structure. Other rRNA post-transcriptional modifications have also been shown to be altered in cancers and attempts have been made to address their functionality. For instance, the m^1^acp^3^ψ modification present at a single site located in the decoding centre of ribosomes was found to be hypomodified in a subset of CRC samples and other cancer types [127]. In colon cancer, m^1^acp^3^ψ hypomodification was associated with increased translation of ribosomal proteins [127]. Another study showed that loss of an rRNA pseudouridylation site reduced survival in a mouse model of HCC [128]. 

Altogether, these studies support the notion that heterogeneous ribosomes exist in certain cancers and may have ‘specialized functions’, whereby they are capable of influencing gene expression via selective translation of mRNAs. Furthermore, structural differences in heterogeneous cancer ribosomes could potentially be exploited for novel therapies [129]. Given the pronounced links between WNT signalling, induction of MYC, and its known role as a master regulator of biogenesis, further exploration of ribosome heterogeneity in certain WNT-driven cancers may open a new therapeutic avenue.

## Figures and Tables

**Figure 1 cells-12-00990-f001:**
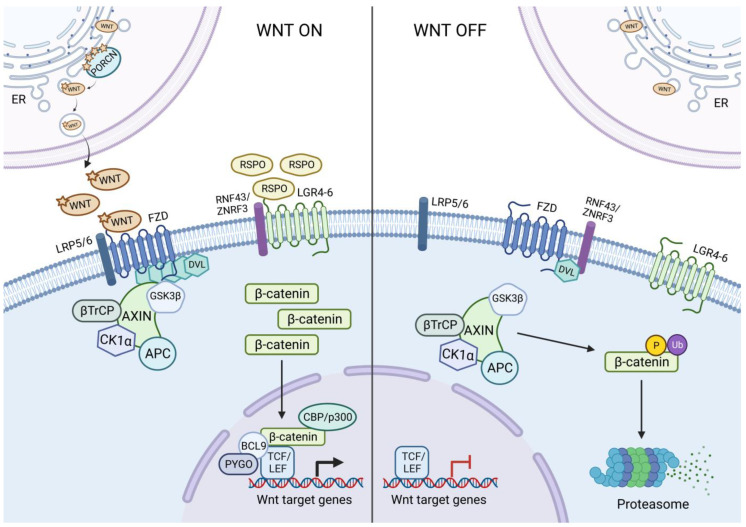
Active and inactive states of canonical WNT signalling. Altogether, numerous feedback loops modulate the activity of the WNT pathway; therefore, perturbation of these feedback pathways or of components of the core canonical WNT pathway can disrupt homeostasis and lead to cancer. Alternatively, these feedback loops could provide potential therapeutic targets. WNT ON: The WNT pathway is switched on (**left**). Porcupine (PORCN) mediates palmitoylation (stars) of WNT ligands at the endoplasmic reticulum (ER) prior to secretion. Modified WNT ligands bind to Frizzled receptors (FZD) and lipoprotein receptor-related proteins 5 and 6 (LRP5/6), then Dishevelled (DVL) polymerizes at the cytosolic side of FZD. The FZD-LRP5/6 receptor complex binds the destruction complex consisting of AXIN, GSK3β, β-TrCP, CK1α, and APC, via AXIN, and thereby allows for the accumulation of newly synthesized β-catenin in the cytosol. Binding of R-spondin (RSPO) to the receptors LGR5 and RNF43/ZNRF3 prevents the inhibition of FZD and potentiates WNT signalling. Following this, β-catenin enters the nucleus and binds to the transcription factors LEF1 and TCF, together with the co-activators BCL9, PYGO, and CBP/p300, initiating the transcription of WNT target genes. WNT OFF: The WNT pathway switched off (**right**). In the absence of extracellular WNT ligands, the FZD-LRP5/6 receptors remain separate and inactive. In addition, absence of RSPO enables RNF43/ZNRF3 to bind and inhibit FZD receptors. The destruction complex is unbound, and its components can phosphorylate and ubiquitinate β-catenin. Hence, β-catenin is recognized and degraded by the proteasome and cannot enter the nucleus to activate transcription. Created with BioRender.com.

**Figure 3 cells-12-00990-f003:**
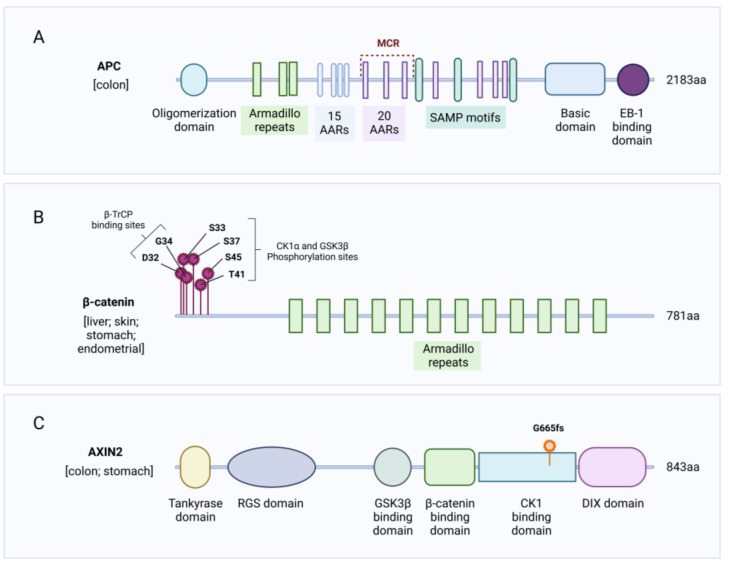
The impact of specific mutations in *APC*, *CTNNB1*, and *AXIN2* on protein functionality in certain cancers. (**A**) Schematic representation of APC protein. 15AARs and 20AARs are required for β-catenin binding, SAMP repeats bind to AXIN. MCR denotes the mutation cluster region of recurrent *APC* mutations found in CRC—these mutations give rise to truncated proteins that lack the SAMP motifs and typically only retain between 1 and 3 intact 20AARs. Therefore, these truncated proteins have significantly reduced β-catenin and AXIN binding capacity. (**B**) Schematic representation of β-catenin protein and recurrent exon 3 mutations found in liver, skin, endometrial, and stomach cancer. Ser33, Ser37, Thr41, and Ser45 are phosphorylated by CK1α and GSK3β marking β-catenin for subsequent ubiquitination. Asp32 and Gly34 are required for correct β-TrCP binding for β-catenin to be ubiquitinated. Point mutations at these 6 amino acids result in the stabilization of β-catenin in the cytoplasm. (**C**) Schematic representation of the AXIN2 protein and location of recurrent frameshift mutation found in colon and stomach cancer. The recurring frameshift mutation at G665 lies within the CK1α binding domain of AXIN2. This mutation induces a truncation and therefore gives rise to a protein with impaired CK1α binding, disrupting the function of the destruction complex. Information on protein domain structures and mutation locations from cBioportal [30,31,38,39]. Figure created with BioRender.com.

**Figure 4 cells-12-00990-f004:**
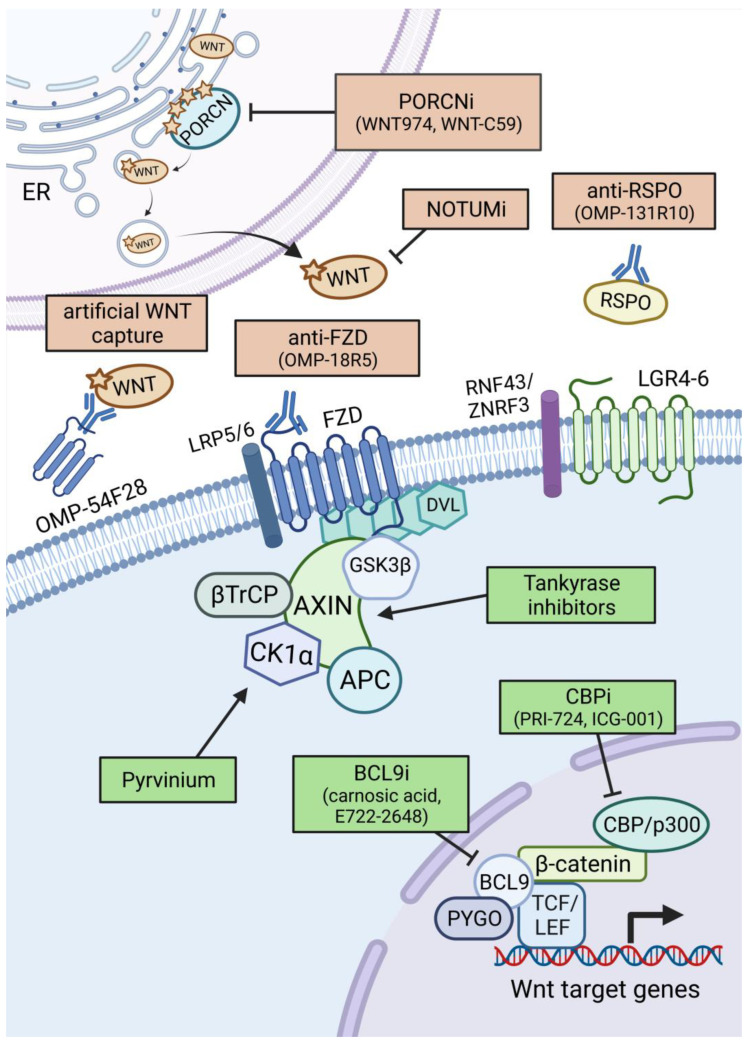
Potential therapies targeting the WNT pathway in cancer. Ligand-dependent (red) inhibitors and antibodies act extracellularly and target WNT ligands and receptors. Ligand-independent (green) inhibitors target β-catenin activity. Ligand-dependent inhibitors are PORCNi (WNT974, WNT-C59) that target the secretion of WNT, the artificial receptor OMP-54F28 that captures secreted WNT with antibodies, anti-FZD (OMP-18R5) antibodies that inhibit FZD receptors, and anti-RSPOs (OMP-131R10) that bind RSPO ligands as well as NOTUMi. Ligand-independent agents are Tankyrase inhibitors that stabilize AXIN and Pyrvinium, which promotes CK1α phosphorylation and results in increased degradation of β-catenin, as well as inhibitors of BCL9 (BCL9i; carnosic acid, E722-2648) and CBP (CBPi; PRI-724, ICG-001) that decrease β-catenin-mediated transcription. Created with BioRender.com.

**Figure 5 cells-12-00990-f005:**
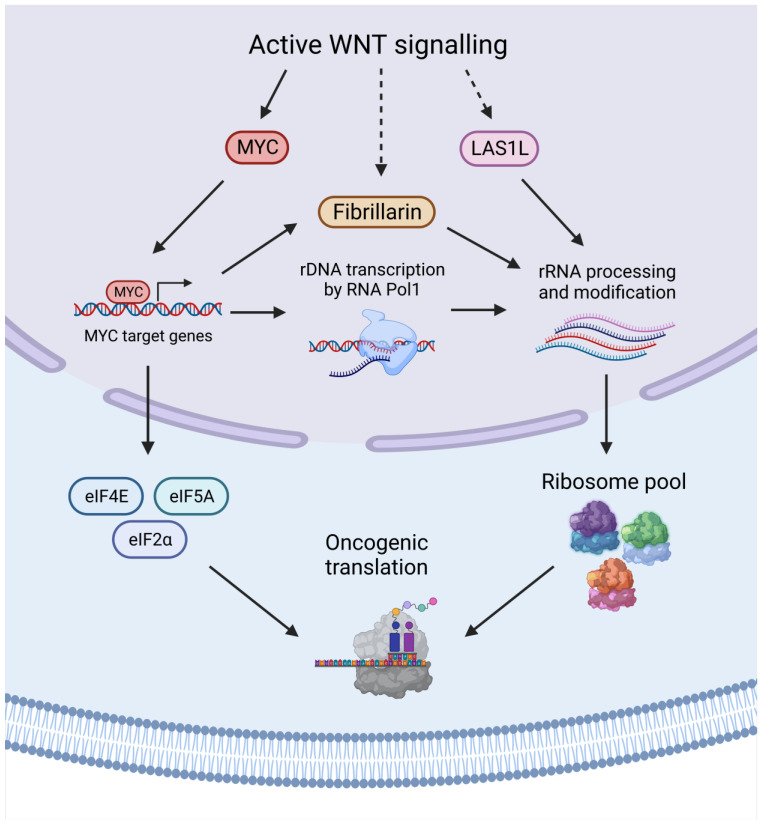
WNT signalling affects ribosome biogenesis and translation. The WNT target genes c-MYC, Fibrillarin, and LAS1L regulate rRNA processing and modification. The transcription factor c-MYC upregulates the transcription of translation initiation factors as well as Fibrillarin and the Polymerase 1 (Pol1)-mediated transcription of rDNA. Differential WNT-driven regulation of ribosome biogenesis may feed the emergence of heterogeneous ribosomes and, together with increased translation factors, potentially affects oncogenic translation. Created with BioRender.com.

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
