# Peer review of "The Role of WNT Pathway Mutations in Cancer Development and an Overview of Therapeutic Options"

_cells, 2023, doi:10.3390/cells12070990_

Round 1

Reviewer 1 Report

The review by Groenewald et al provides a comprehensive summary of the available literature on the role of WNT and associated signaling pathways in cancer development, and the therapeutic approaches taken to target these pathways in various cancers. Most of the discussions are focused on CRC but the authors have included data from other cancers as well such as HCC, PDAC etc. Overall, it is a very well written review, and this reviewer does not have any major comments; couple of minor suggestions are provided below.

Minor comments:

Line 92: change 'activity of the WNT pathway activity' to 'activity of the WNT pathway'.

Line 120: Change 'figure 2' to 'Figure 2'

Line 175: Change 'ubiquitination of is impaired.' to 'ubiquitination is impaired.'

Line 257: Sentence starting 'Whilst....', it appears to be incomplete, please combine this with the sentence that follows this or rephrase.

Author Response

We thank the reviewer for these comments and have amended the text accordingly.

Reviewer 2 Report

The topic of the review is interesting, however, some modifications are required.

1)      I seek what is new knowledge in this review and found there is no novelty. Moreover, recently some comprehensive reviews have been published in Wnt Signaling. Such as: https://www.nature.com/articles/s41568-020-00307-z

2)      Wnt signaling is an important pathway for immune cell maintenance and renewal. However, the interaction between Wnt pathway genes and the immune system has not been mentioned.

3)      There 19 genes have been recognized in Wnt signaling. The authors must list them in a table before figure 2 and write the function of each gene in that table briefly.

4)      Usually when the word “ mutation” is applied, it referred to changes in DNA sequence. However, the Wnt signaling pathway is regulated by epigenetic marks. On the other hand, the authors must clarify the aberration of the genes in Wnt signaling contributes to cancers caused by changes in DNA sequence or by epigenetic aberration. 

Author Response

The topic of the review is interesting, however, some modifications are required.

1)      I seek what is new knowledge in this review and found there is no novelty. Moreover, recently some comprehensive reviews have been published in Wnt Signaling. Such as: https://www.nature.com/articles/s41568-020-00307-z

We politely disagree with the reviewer over this comment. There is a plethora of existing WNT signalling reviews already out there, so we are operating in a very crowded field. However, we feel that the broad nature of our review encompassing multiple mutations, multiple cancers along with targeting therapies provides a solid foundation to individuals finding their way in the field, without going into as much specific detail as other reviews. For instance, the review listed above has a strong focus on AXIN mutations, but only very briefly mentions RSPON mutations.

Finally, we believe our final section describing deregulated protein synthesis downstream of WNT signalling in several cancers is highly novel in a broad WNT signalling in cancer review.

2)      Wnt signaling is an important pathway for immune cell maintenance and renewal. However, the interaction between Wnt pathway genes and the immune system has not been mentioned.

Whilst the interaction of the WNT pathway and immune cell maintenance and renewal are very interesting, we feel this is beyond the scope of our review and as such have not focused on this area.

3)      There 19 genes have been recognized in Wnt signaling. The authors must list them in a table before figure 2 and write the function of each gene in that table briefly.

We politely disagree with this suggestion since we feel that our introduction into the WNT pathway and accompanying figure on the WNT canonical pathway introduces the key players in this pathway along with their functions.

4)      Usually when the word “ mutation” is applied, it referred to changes in DNA sequence. However, the Wnt signaling pathway is regulated by epigenetic marks. On the other hand, the authors must clarify the aberration of the genes in Wnt signaling contributes to cancers caused by changes in DNA sequence or by epigenetic aberration. 

We thank the reviewer for raising the important point of epigenetic regulation of WNT pathway components in cancer. It is true that we have only focused on DNA mutations, we have added in a sentence stating the role of epigenetic regulation of the WNT pathway pointing our readers in the direction of several studies in this important area. Nonetheless we feel that this area is beyond the scope of our review.

Moreover, in agreement with reviewer 3, we have added in a figure showing how certain hotspot mutations in different genes can alter the protein function of specific WNT pathway components. -Further placing the emphasis on DNA mutations, as opposed to epigenetics.

Reviewer 3 Report

In this manuscript, the authors reviewed the predominant driving mutations found in cancer that lead to WNT pathway activation. Also, the authors summarized some of the therapeutic interventions currently available against tumors harboring either ligand-dependent or -independent mutations. The authors also discussed a potential new therapeutic avenue by targeting translational apparatus downstream from WNT signaling.

The authors provided sufficient background information in the review. The language used in the manuscript is clear and professional.

However, the authors mentioned various mutations in section 2 " Wnt activating mutations in cancer". It would be more clear to the readers if the authors can show illustrations with the locations of the mutation in the protein and which domains are related. 

Other than that, the manuscript is very well-written.

Author Response

In this manuscript, the authors reviewed the predominant driving mutations found in cancer that lead to WNT pathway activation. Also, the authors summarized some of the therapeutic interventions currently available against tumors harboring either ligand-dependent or -independent mutations. The authors also discussed a potential new therapeutic avenue by targeting translational apparatus downstream from WNT signaling.

The authors provided sufficient background information in the review. The language used in the manuscript is clear and professional.

However, the authors mentioned various mutations in section 2 " Wnt activating mutations in cancer". It would be more clear to the readers if the authors can show illustrations with the locations of the mutation in the protein and which domains are related. 

We thank the reviewer for this suggestion. It was something that we had also originally considered. We have added in a new figure (new figure 3) showing how certain recurrent hotspot mutations in APC, CTNNB1 and AXIN2 identified in certain cancers affect particular protein domains that are essential for that protein’s function.

Other than that, the manuscript is very well-written

Round 2

Reviewer 2 Report

The authors answered my comments and I would recommend publishing.